# The impact of Worms and Ladders, an innovative health educational board game on Soil-Transmitted Helminthiasis control in Abeokuta, Southwest Nigeria

Dorcas B. Bassey[1], Hammed O. Mogaji[2]*, Gabriel A. Dedeke[1], Bolanle I. Akeredolu-Ale[3], Eniola M. Abe[4], Akinola S. Oluwole[5], Abdulhakeem A. Adeniran[1,6], Olagunju A. Agboola[1,7], Chiedu F. Mafiana[8], Uwem F. Ekpo[1]

1 Department of Pure and Applied Zoology, Federal University of Agriculture, Abeokuta, Ogun State, Nigeria, 2 Department of Animal and Environmental Biology, Federal University Oye-Ekiti, Ekiti State, Nigeria, 3 Department of Communication and General Studies, Federal University of Agriculture, Abeokuta, Ogun State, Nigeria, 4 National Institute of Parasitic Disease and Control, China Centre for Disease Control, People's Republic of China, 5 COUNTDOWN project, Sightsavers, Nigeria Country Office, Nigeria, 6 Laboratory of Molecular Biomedicine, Centre for Genomic, Biotechnology, Instituto Politecnico Nacional, Mexico, 7 Department of Biological Sciences, Lead City University, Ibadan, Nigeria, 8 Directorate of Research and Innovation, National Open University of Nigeria, Abuja, Nigeria

* mogajihammed@gmail.com

## Abstract

In most endemic sub-Saharan African countries, repeated infections with soil-transmitted helminth (STH) occur as early as six weeks after the end of mass drug administration (MDA) with albendazole. In this study, we designed a new health educational board game Worms and Ladders and evaluated its potential to complement MDA with albendazole and reduce reinfection rates through the promotion of good hygiene practices among school-aged children. The evaluation employed a randomized control trial (RCT) design. Baseline knowledge, attitude and practices (KAP) relating to STH were obtained using a questionnaire from 372 pupils across six schools in Abeokuta, Nigeria. Schools were randomly assigned into intervention and control group, with the former and latter receiving Worms and Ladders and the common Snake and Ladder board game respectively. Fresh stool samples were also collected at baseline for STH diagnosis before administering 400mg single dose albendazole. Follow-up assessments of STH burden and KAP were conducted three and six months' post-intervention. Data generated from the study were analyzed using SPSS 20.0 software, with confidence interval set at 95%. Prevalence of STH dropped from 25.0% to 10.4% in the intervention group and 49.4% to 33.3% in the control group at three months' post-intervention. The prevalence further dropped to 5.6% in the intervention group at six months' post-intervention. However, it increased to 37.2% in the control group at six months' post-intervention. There was a significant difference (p<0.05) in prevalence after intervention among the groups. KAP on transmission, control and prevention of STH significantly improved (p<0.05) from 5.2% to 97.9% in the intervention group compared to 6.2% to 7.1% in the control group. The Worms and Ladders board game shows the potential to teach and promote good hygiene behavior among SAC. These findings posit the newly developed

**Data Availability Statement:** All relevant data are within the manuscript and its Supporting Information files.

**Funding:** The author(s) received no specific funding for this work.

**Competing interests:** The authors have declared that no competing interests exist.

game as a reliable tool to complement mass drug administration campaigns for STH control.

## Author summary

School-aged children are the most affected group of people in terms of burden due to soil-transmitted helminth infections. Unfortunately, the available treatment programme with albendazole cannot prevent reinfection. Health and hygiene education has been recommended to be effective at reducing the rate of STH infections through increased knowledge about transmission and improved hygiene attitude and practices. We, therefore, developed a health educational board game Worms and Ladders and evaluated its potential to complement treatment and reduce reinfection rates. Our findings show that the worm burden dropped significantly among children who played the newly developed game, compared to other children who played another game. The knowledge, attitude and practices of the children as regards STH also improved significantly. The Worms and Ladders board game, therefore, has the potential to promote good hygiene behavior, which in turn translated to a reduced rate of infections. These findings present the newly developed game as a reliable tool to complement mass drug administration campaigns for STH control.

## Introduction

Soil-Transmitted Helminthiasis (STH) caused by *Ascaris lumbricoides*, *Trichuris trichiura*, *Necator americanus* and *Ancyclostoma duodenale* is one of the world's most serious public health problems [1–4]. Over 600 million school-aged children live in areas where these parasites are intensively transmitted. About 24% of the world's population (>1.5 billion people) are infected, with the highest numbers occurring in sub-Saharan Africa, the Americas, China and East Asia [5].

The current strategy for control of morbidity is the periodic treatment of school-aged children living in endemic areas where poor water, sanitation and poor personal and domestic hygiene prevail [6]. The World Health Organization recommends mass administration of anthelmintic drugs (albendazole/mebendazole) to at-risk populations, either once a year (annually) when the baseline prevalence of infections in the community is over 20% or twice a year (biannually) when the prevalence is over 50% [7].

Despite successes documented with mass administration campaigns for anthelminthic drugs, an upsurge in prevalence and reinfection rates after treatments is not uncommon [8,9]. Reinfection often occurs after deworming is ceased and progresses to pre-treatment levels in as few as six weeks and up to 94% at the twelfth month [10–12]. Although mass drug administration is the cornerstone of infection control, it is no longer news that the approach cannot prevent reinfection and thus calls for the need of additional public health measures, such as health education, to complement and sustain achieved efforts [13]. Health education is any combination of learning experiences designed to help individuals or communities improve their health by increasing their knowledge or influencing their activities [14]. The health education strategy is economical as it reduces the cost of deworming, increases the level of overall health knowledge and acceptability of deworming interventions [15,16]. It also enhances increased understanding of effective health practices [17]. The inclusion of health education

into school-based deworming programmes ensured that exposure and transmission to STH infections are reduced, through improved knowledge and behavioral changes [18,19]. There is a plethora of evidence reinforcing the theory above on the significant role health educational activities have on positive behavioral change [20–23]. For instance, Bieri and colleagues [13] in China used a video-based health education package to prevent new STH infections after treatment. Similarly, Ejike and colleagues [24] in Nigeria has used the health education game called Schisto and Laddders to significantly improve the knowledge, attitudes and practices of school children for the prevention and control of schistosomiasis. Health education thus plays a significant role in helminths disease prevention, by decreasing exposure risk through behavioral change, and in increasing health-seeking behavior.

There are over 5 million school-aged children predicted to be infected with STH in Nigeria [25]. The accrued setbacks of rising reinfection patterns after treatment, there is a need to develop and invest in new health educational tools to complement the administration of medicines in the control of these infections. In this study, we designed a health educational board game Worms and Ladders, and evaluated its potential for the control of STH among SAC, though the promotion of positive behavioural changes.

## Methods

### Ethics statement

Ethical clearance for this study (SUBEB/SS/1102) was obtained from Ogun State Universal Basic Education Board (SUBEB) health ethics review board. Pre-survey contact/advocacy meeting was made to each selected school to obtain consents from the headteacher after explaining the objectives of the research to them. Schools willing to participate in the study completed written consent forms at the initial stage. Subsequently, parents and guardians/caregivers were also informed about the study through the Parents-Teachers-Association (PTA) meeting organized by the consenting schools. Parents were asked to provide parental consents to their children by completing another consent form. However, an assent form was completed in cases where the child/ward is below 16 years of age. Children whose parent did not agree to the study procedures were not invited to participate in the research. However, all other children were invited and voluntarily asked to participate in the study irrespective of their parent's decision. They were informed of the study procedures, including the need for them to be available over 6 months for follow up. Only children who voluntarily agreed to participate at this stage were recruited into the study. The method of consent assertion was through thumbprint on already printed informed consent forms (ICFs).

### Study area and study participants

The study was undertaken in six public primary schools located in Abeokuta, the capital city of Ogun State, Nigeria. The area lies in the rain forest vegetation belt of the country on co-ordinates 7°09′00″N 3°21′00″E. It is highly urbanized, majorly because it is the capital of the State. Studies have shown that Abeokuta is endemic for STH with high prevalence among SAC [10, 26–28]. Study participants were SAC (5-15years old) with no history of deworming in the past six months.

### Study design and selection of schools

This study used a Randomized Control Trial (RCT) design involving six (6) public primary schools selected out of the 49 public primary schools in the study area. As an initial step, the forty-nine schools in the study area were stratified into two clusters (Cluster 1 and Cluster 2)

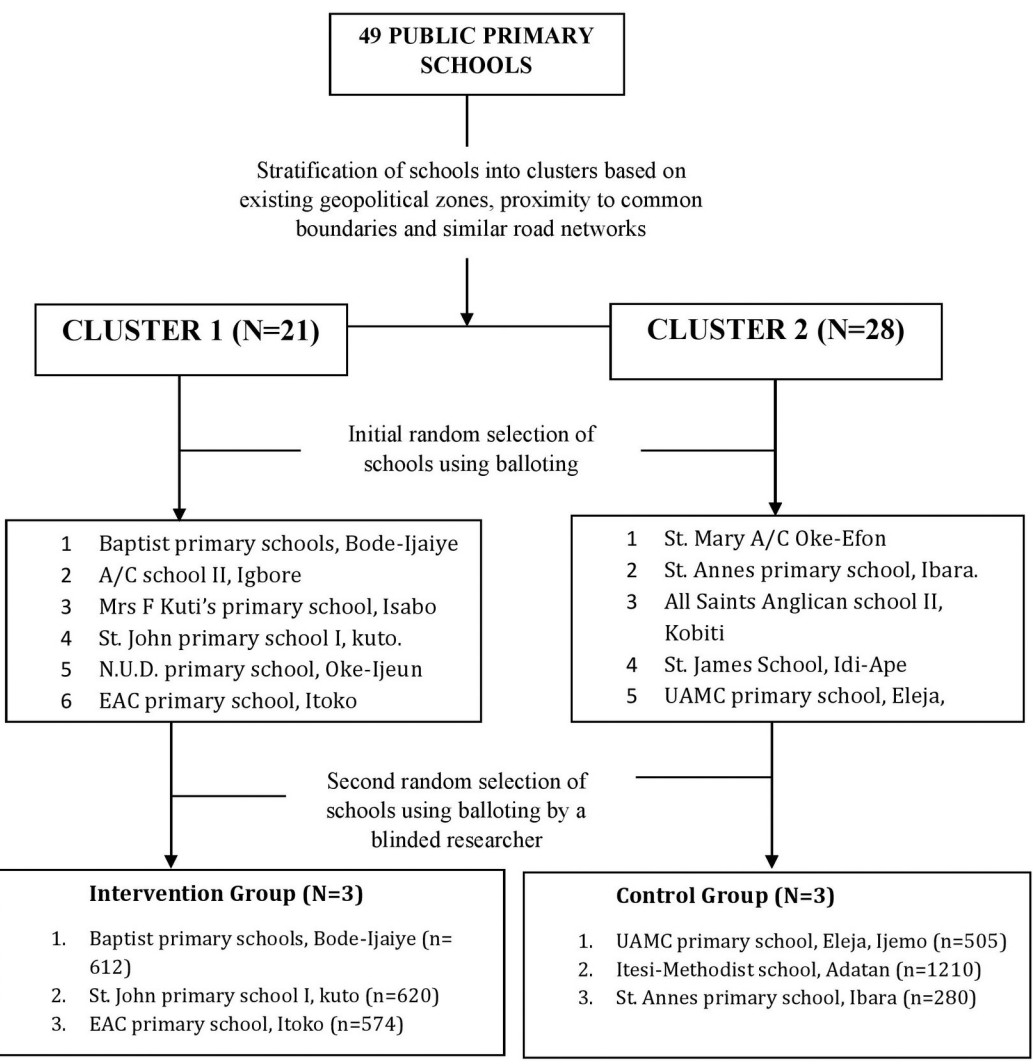

**Fig 1. Flow chart on school selection procedures.**

based on existing geo-political zones, proximities to common boundaries and road networks (Fig 1). Prior to commencement of schools selection, the newly designed worms and ladders game (intervention) and the conventional snake and ladder game (control) were randomly assigned to Cluster 1 (C1) and Cluster 2 (C2) respectively. The intervention schools were then selected from C1 with a total of 21 schools; and the control schools were selected from C2 with a total of 28 schools. In order to have an ample sampling frame, six schools were randomly selected, at first using balloting from each cluster. Afterwards, three schools were selected randomly out of the six schools in each clusters using the second round of balloting to minimize the cofounding effect that may arise due to cross-communication among schools that shares boundary or are near each other [24]. The final randomization of participating schools to intervention was carried out by one of the researchers (blinded) who did not partake in the schools' selection procedures. Preliminary visits were made to selected schools to document cofounding factors such as the presence of UNICEF-assisted school-based WASH intervention programmes. Schools with UNICEF-assisted programme were replaced purposively with one of the three remaining schools in the sampling frame. The schools in cluster 1 (intervention

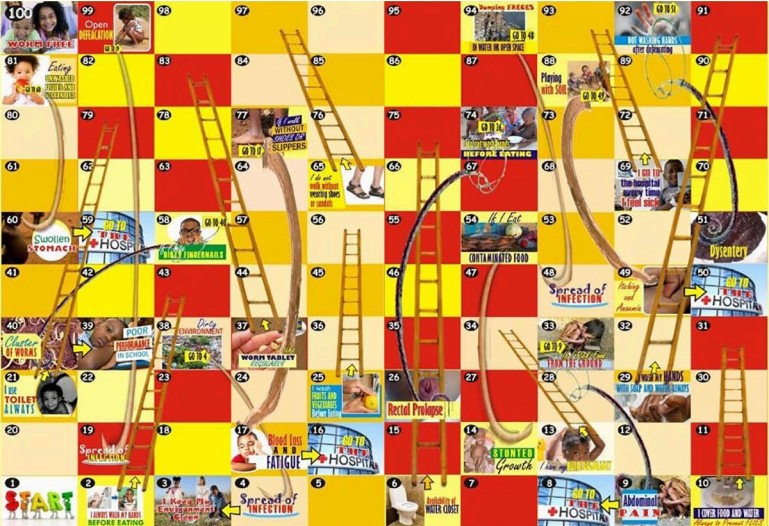

**Fig 2. The newly designed Worms and Ladders Game.**

group) received the newly designed Worms and Ladders game (Fig 2) while those in cluster 2 (control group) received the common Snakes and Ladders game (Fig 3). The study was conducted from March to October 2015, and it involved five distinct phases; which are baseline, deworming, intervention, three and six-month post-intervention follow-up phases.

## Sample size determination and selection of study participants

The sample size for this study was determined using the formula; $n_s = \frac{N}{1+Ne^2}$, as described by Yamane [29], where $n_s$ = sample size, N = total study population and e = marginal error. As an initial step, enrollment registers for selected schools were reviewed to obtain the total number of school-aged children. The enrollment figure from selected schools was summed up to obtain a total population (N) of 3801 school-aged children. In determining the sample size, a

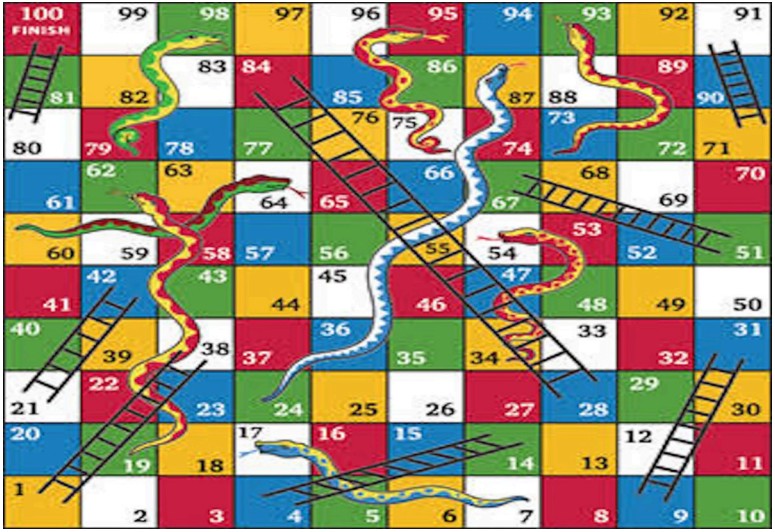

**Fig 3. The common Snake and Ladder Game.**

marginal error (e) of 5% was considered at 95% level of confidence. The minimum sample size determined, therefore, was 362, i.e. an average of 60 school-aged children per selected school. Selection of pupils in each school was carried out after stratification by class grade from Primary 1 to 6. Children from each class grade were voluntarily invited to participate, and only those who consented were recruited into the study. Finally, a total of 372 school-aged children were recruited from the six schools. However, in most schools the number of consenting children varied by class grade hence giving roughly an unequal number of pupils recruited.

## Designing the Worms and Ladders game

As an initial step during the development of an intervention tool, a consultation meeting among investigators was held to identify risk factors for STH infections. The risk factors for STH were considered through desk-review of previous epidemiological studies conducted among school-aged children. Since studies have shown that poor sanitary conditions and personal hygiene such as not wearing shoes or slippers, indiscriminate defecation, not washing hands before eating and/or after using the toilet, lack of toilet facilities and dirty environment predisposes children to STH infections [10, 30–32]. The health educational messages were then formed in response to each of the identified risk factor. The messages were presented in simple forms to ensure proper understanding. Ten major health educational messages developed were; washing of hands before eating, washing hands with soap after using the toilet, not playing with soil, wearing of slippers or shoes when going outside, avoiding open (indiscriminate) defecation, washing of fruits and vegetables before consumption, covering of food from flies, keeping fingernails clean and cut, not picking food from the ground, and cleaning the environment.

Furthermore, since the target population for the tool is school-aged children, the team adopted the use of visual aids, as it is one of the strongest tools used in communicating messages; particularly when accompanied with interactive methods [33]. Also, the team considered deploying this tool in a way that facilitates group discussion and demonstration. In consideration of all these, a board game was selected as a means to deliver the health education messages to the school children. Therefore, a board game named Worms and Ladders was developed to teach the school-aged children about transmission, symptoms, treatment, prevention and control of STH infections. The game was piloted in a school setting to identify areas that need to be worked upon to enhance clarity and facilitate understanding. The pilot study was implemented in an area different from where the study schools were selected.

The design of the Worms and Ladders health educational board game follows the popular Snakes and Ladders game, based on the concept of reward for good health behavior by moving up a ladder and punishment for risky health behavior by being bitten by the STH worms. Graphic design was done in CorelDraw X13 software. Illustrated pictorials, which explain the characters were used in the design of the game for easy understanding and relativity even to a layman, thus improving the all-round development and the intellect of the average child. The game consists of squares numbered from 1 to 100. Play starts from square number one and ends at the hundredth square, which is worm free. The anterior (head) end of the worms are placed on squares inscribed with statements such as unhygienic behaviors, poor sanitary conditions, which predispose school children to STH infections and transmission. However, the posterior (tail) end of the worm falls on statements that depict the consequences of such unhygienic behavior or sanitary conditions.

On the other hand, the foot of the ladders is placed strategically on the hygienic behaviors (preventive measures) that prevent infection STH. The game can be played by a minimum of 2 and a maximum of 4 persons, to allow proper interaction and assimilation of the health

education messages and warnings on the board, using dice and counters. The game starts when a player throws a 1 with a dice. A Player moves up when he/she gets to the foot of a ladder and goes down when he gets to the head of the STH worm. Players are self-guided by the health messages on each square of the board. The players count the number shown on the dice when thrown and the first player to get to the square marked 100 wins the game.

## Baseline, follow-up and end-line assessments of STH infections and Knowledge, Attitude and Practices of the pupils

A cross-sectional approach was employed for the collection of fecal samples. Fecal samples were collected in a single day at each school at the following timelines: baseline, follow-up (3 months) and end line (6-months). One gram of fecal sample was taken from the samples provided by study participants. The fecal samples were emulsified in 10ml of Sodium Acetate Acetic Acid Formalin (SAF) solution and transported to the laboratory. Samples were further processed using ether-concentration method in the laboratory and examined under the microscope for ova of STH parasites [34].

Children's knowledge, attitude and practices (KAP) towards STH infections were also assessed using structured questionnaires and focus group discussions on three different occasions during the lifetime of the study, i.e. baseline, follow-up (3 months) and end line (6-months). The children were interviewed individually in the headteachers' office, within the school premises. Where necessary, the interviews were conducted in the native Yoruba language. Following baseline assessments, all the children, including the selected participants in the selected schools, were dewormed with 400mg albendazole (single dose). Deworming was carried out by State STH control programmers according to WHO guideline. To verify that the selected study participants were not infected, egg-output were monitored for six days after deworming. Only non-infected children in intervention and control schools were allowed to participate further in the intervention phase of the study. The designed health education game Worms and Ladders was given to children in the intervention group.

In contrast, the Snakes and Ladders game was given to children in the control group. Both categories of children were trained on how to use (play) the game during leisure hours under the supervision of a teacher and one of the researcher. Supervisory weekly visits were made to each group through the lifetime of the study. Following the deworming at baseline, children were monitored for 6 months. They did not take part in any deworming exercise throughout the study period. Follow-up cross-sectional assessment of STH infection status and KAP was done 3-months post-baseline after the game was introduced, and end-line assessments were carried out in another 3-month (6 months post-baseline). All study participants were dewormed again after end line assessments using 400mg albendazole (single dose).

## Evaluation of the Worms and Ladders game

The pupils played the Worms and Ladders board game in the intervention schools. The pupils were re-oriented on the need to work with their teachers as they train them on how to play the game. As an initial step, the researchers trained class teachers on how to play the game and how to monitor the playing of the game. After that, the game was introduced to the pupils, and they were trained on how to play the game. The decision to play the game during leisure hours was reached afterwards between the school management, teachers and pupils. Weekly visits were made to the intervention schools to monitor and discuss with the pupils whenever they are playing the game. About 40 minutes of the leisure period was expended on playing the game every school day. At the end of the leisure periods, the games were returned to the supervising teachers for safekeeping till the next school day.

## Group Discussion (GD)

During the monitoring and follow-up period, players were invited to participate in a group discussion on the Worms and Ladders game. Discussions were among a mixed group of boys and girls and were conducted during and after playing the games. All discussions, including the impact of the game of players attitude and behavior, players understanding, and interpretation of the health education messages and symbols were recorded, coded and extracted under different themes during analysis. The discussants were informed about the recordings and encouraged to be as free as possible.

## Data tabulations and statistical analysis

Raw data were entered into Excel database (S1 Data), and then analyzed using SPSS IBM 20.0 version, Armonk, NY, IBM Corp. Data were first subjected to descriptive statistics including frequencies and cross-tabulations, followed by Pearson chi-square analysis to test for significant differences in proportion (impact of the designed health education game) on prevalence, and KAP of children at baseline, follow up and end line visits. Logistic regression model was used to analyse the longitudinal data. Odds ratio of infection at baseline, three months and six months follow-up was calculated using EpiInfo version 5. Independent sample T-test was also used ascertain the difference in mean intensity of STH infection among participants in the intervention and control group, respectively. The confidence interval was set at 95%.

# Results

## Demographic characteristics of study participants

A total of 372 children, 190 (51.1%) males and 182 (48.9%) females between the age group 5–10 years (197; 53%) and 11-15years (175; 47%) participated in this study with 212 (56.9%) in the intervention group and 160 (49.1%) in the control group (Table 1).

An overall STH prevalence of (25.0% vs 10.4%) and (49.4% vs 33.3%) was recorded for baseline versus follow-up assessments in the intervention and control group, respectively. However, for end line assessment, prevalence dropped for the intervention group to 5.6% but increased to 37.2% for the control group. There were significant differences in the reduction rate for overall STH infection across the two follow-up periods (p = 0.007). The odds ratios also reduced progressively from baseline to end-line assessment in the intervention group as

**Table 1. Demographic characteristics of study participants.**

| Group | Intervention group NE (%) | Control group NE (%) | Total (%) |
|---|---|---|---|
| **Sex** | | | |
| **Male** | 117(31.5) | 95(25.5) | 190(51.1) |
| **Female** | 73(19.6) | 87(23.4) | 182(48.9) |
| **Total** | 212(51.1) | 160(48.9) | 372(100) |
| **p-value** | 0.067 | | |
| **Age group (years)** | | | |
| **5–10** | 101(27.2) | 96(25.8) | 197(53) |
| **11–15** | 111(29.8) | 64(17.2) | 175(47) |
| **Total** | 212(51.1) | 160(48.9) | 372(100) |
| **p-value** | 0.018 | | |

NE: Number Examined

STH infection prevalence at baseline, follow-up and end-line assessment

**Table 2. Prevalence of STH infections and risk differences between intervention and control group at baseline, follow-up and end-line assessment.**

| | | Baseline | | Follow-up (3-months) | | Endline (6-months) | | p-value |
|---|---|---|---|---|---|---|---|---|
| | | Int. | Con. | Int. | Con. | Int. | Con. | |
| | | N = 212 | N = 160 | N = 193 | N = 135 | N = 142 | N = 113 | |
| STH | No. of Positives (%) | 53 (25.0) | 79(49.4) | 20 (10.4) | 45(33.3) | 8 (5.6) | 42 (37.2) | 0.007 |
| | No. of Negatives (%) | 159(75.0) | 81 (50.6) | 173 (89.6) | 90(66.7) | 134 (94.4) | 71 (62.8) | |
| | p-value | <0.00001 | | <0.00001 | | <0.00001 | | |
| | OR (95% CI) | 0.34(0.22–0.53) | | 0.23(0.13–0.41) | | 0.10 (0.04–0.23) | | |
| *Ascaris spp.* | No. of Positives (%) | 49 (23.1) | 66 (41.2) | 15 (7.8) | 40(29.6) | 6 (4.2) | 42 (37.2) | 0.001 |
| | No. of Negatives (%) | 163 (76.9) | 94 (58.8) | 178 (92.2) | 95(70.4) | 136 (95.8) | 71 (62.8) | |
| | p-value | 0.00018 | | <0.00001 | | <0.00001 | | |
| | OR (95% CI) | 0.43(0.27–0.67) | | 0.20(0.10–0.38) | | 0.07(0.03–0.18) | | |
| Hookworm | No. of Positives (%) | 22 (10.4) | 31 (19.4) | 2(1.0) | 4(3.0) | 1 (0.7) | 2 (1.8) | |
| | No. of Negatives (%) | 190 (89.6) | 129 (80.6) | 191(99) | 131(97.0) | 141 (99.3) | 111(98.2) | 0.898 |
| | p-value | 0.013 | | 0.20046 | | 0.4330 | | |
| | OR (95% CI) | 0.48(0.27–0.87) | | 0.34(0.06–1.89) | | 0.39 (0.03–4.39) | | |
| *Trichuris spp.* | No. of Positives (%) | 6 (2.8) | 6 (3.8) | 3(1.6) | 3(2.2) | 2 (1.4) | 0(0) | |
| | No. of Negatives (%) | 206 (97.2) | 154 (96.2) | 190(98.4) | 132(97.8) | 140 (98.6) | 113(100) | - |
| | p-value | 0.619 | | 0.611 | | - | | |
| | OR (95% CI) | 0.75(0.24–2.36) | | 0.66 (0.13–3.33) | | - | | |

Int.: Intervention group; Con: Control group; OR: Odd ratios; CI: Confidence interval

compared to the control group (Table 2). By species, *Ascaris lumbricoides* prevalence reduced from 23.1% at baseline to 7.8% at the first follow-up and further reduced to 4.2% during the end line assessment in the intervention group (Table 2). However, in the control group, *Ascaris lumbricoides* prevalence decreased from 41.2% at baseline to 29.6% at the first follow-up. It increased to 37.2% at the second follow-up. There were significant differences in the reduction rate for *Ascaris lumbricoides* across the two follow-up periods (p = 0.001). The odds ratios also reduced progressively from baseline to end line assessment in the intervention group as compared to the control group.

Similarly, hookworm prevalence decreased from 10.4% at baseline to 1.0% at the first follow-up and 0.7% at second follow-up in the intervention group (Table 2). However, the prevalence decreased from 19.4% at baseline to 3.0% at first follow-up and further reduced to 1.8% at second follow-up in the control group. There were no significant differences in the reduction observed for hookworm infection across the two follow-up periods (p = 0.898). The odds ratios for hookworm in the intervention group also reduced at first follow-up. However, they increased to a sub-baseline level at second-follow up. *Trichuris trichiura* infection also decreased from 2.8% at baseline to 2.1% at first follow up and 1.4% at second follow-up in the intervention group. However, in the control group, the prevalence also decreased from 3.8% at baseline to 2.2% at the first follow-up, and no infection was recorded during the end line assessment. The odds ratios for *Trichuris trichiura* infection also reduced at first follow-up in the intervention group.

## Mean Intensity of STH infection at baseline, follow-up and end-line assessment

Generally, there was a reduction in mean intensities of STH infection through the three phases of assessment. However, only *Ascaris lumbricoides* had a sharp, significant decrease (p<0.001) in intensities between the intervention and control group (Table 3).

**Table 3.** *Mean Intensity of STH infection at baseline, follow-up and endline assessment.*

| STH *species* | Hookworm Mean EPG±SE | | | A. *lumbricoides* Mean EPG±SE | | | T. *trichiura* Mean EPG±SE | | |
|---|---|---|---|---|---|---|---|---|---|
| | Int. | Con. | p-value | Int. | Con. | p-value | Int. | Con. | p-value |
| Baseline | 0.1074±0.0269 | 0.6731±0.0766 | 0.008 | 0.4778±0.0645 | 0.6731±0.0766 | 0.184 | 0.0221±0.0103 | 0.0230±0.011 | 0.925 |
| Follow-up (3 months) | 0.0094±0.0074 | 0.0249±0.0149 | 0.045 | 0.0946±0.0254 | 0.3303±0.0504 | <0.001 | 0.0164±0.0094 | 0.0119±0.008 | 0.487 |
| Endline (6 months) | 0.0021±0.0021 | 0.0053±0.0038 | 0.118 | 0.0550±0.0234 | 0.4372±0.0612 | <0.001 | 0.0055±0.0040 | 0.0000±0.000 | 0.013 |

Int.: Intervention group; Con: Control group; EPG; egg per gram; SE: standard error

## Impact of Worms and Ladders on the knowledge of the school children about the transmission of STH infection

There was no significant difference in the knowledge of the school children in the intervention and control group about the transmission of STH at baseline. However, three and six months after playing the Worms and Ladders game, there was a significant difference in the pupils' knowledge about transmission, prevention, and control of STH infections in the intervention group compared to the control group (p<0.001). Almost all (98.4% at three months after MDA assessment and 97.9% at six months after MDA) the school children in the intervention group knew at least one way of acquiring STH infections compared to their counterparts in the control group (9.6% at three months after MDA and 7.1% at six months after MDA). A higher percentage of the school children in the intervention group were more knowledgeable about the transmission of STH while the children in the control group still had little or no knowledge about transmission of STH (Table 4).

## Impact of Worms and Ladders on the knowledge of the school children about prevention and control of STH infection

Knowledge of the school children towards prevention and control of STH was not significantly different (P>0.05) among the intervention and the control groups at baseline. The wearing of shoes/slippers, use of toilet/availability of water closet, not picking food from the ground among others were neither mentioned by the intervention group nor the control group as part

**Table 4. Knowledge about transmission of STH among the school children.**

| | Int n = 212 | Cont. n = 160 | p- value | Int n = 193 | Cont. n = 135 | p- value | Int n = 142 | Cont. n = 113 | p- value |
|---|---|---|---|---|---|---|---|---|---|
| **How can we acquire STH infection** | N (%) | N (%) | | N (%) | N (%) | | N (%) | N (%) | |
| Know at least one way of transmission | 11 (5.2) | 10 (6.2) | 0.661 | 190 (98.4) | 13 (9.6) | <0.001 | 139 (97.9) | 8 (7.1) | <0.001 |
| Playing with soil | 4 (1.9) | 2 (1.2) | 0.629 | 77 (39.9) | 6 (4.4) | <0.001 | 58 (40.8) | 2 (1.8) | <0.001 |
| Walking barefoot | 3 (1.4) | 2 (1.2) | 0.891 | 109 (56.5) | 1 (0.7) | <0.001 | 74 (52.1) | 1 (0.9) | <0.001 |
| Not washing hand before eating | 2 (0.9) | 3 (1.9) | 0.440 | 63 (32.6) | 4 (3.0) | <0.001 | 46 (32.4) | 3 (2.7) | <0.001 |
| Open defecation | 0 | 0 | - | 94 (48.7) | 0 | <0.001 | 77 (54.2) | 0 | <0.001 |
| Eating contaminated food | 1 (0.5) | 1 (0.6) | 0.841 | 16 (8.3) | 1 (0.7) | 0.002 | 6 (4.2) | 0 | 0.027 |
| Biting fingernails | 0 | 0 | - | 37 (19.2) | 0 | <0.001 | 26 (18.3) | 0 | <0.001 |
| Not washing hand after defecation | 0 | 0 | - | 49 (25.4) | 0 | <0.001 | 40 (28.2) | 0 | <0.001 |
| Eating unwashed fruits and vegetables | 0 | 2 (1.2) | 0.103 | 41 (21.2) | 1 (0.7) | <0.001 | 34 (23.9) | 2 (1.8) | <0.001 |
| Picking food from ground | 0 | 0 | - | 20 (10.4) | 0 | <0.001 | 14 (9.9) | 0 | 0.001 |
| Not covering food | 1 (0.5) | 0 | 0.384 | 16 (8.3) | 0 | 0.001 | 11 (7.7) | 0 | 0.002 |

* Int- Intervention group, Cont.- Control group, n- Number

of prevention and control of STH. However, three and six months after intervention with Worms and Ladders game, there was a significant increase in the knowledge of the pupils about prevention and control of STH in the intervention group compared to the control group (P<0.001). 96.9% of the school children in the intervention group know at least one way of preventing STH three months after Worms and Ladders game intervention compared to 5.7% at baseline. Handwashing practices and the wearing of shoes have the highest frequency of 37.3% and 40.4% respectively at three months after MDA. Six months after Worms and Ladders game intervention, hand washing practices and the wearing of shoes were mentioned by 54.2% and 52.1% of the school children in the intervention school (Table 5).

### Knowledge about Signs and Symptoms of STH among the school children

At baseline, there was no significant difference in the knowledge about signs and symptoms of STH infections among the intervention and control groups except for abdominal pain and a swollen stomach. However, at three months and six months after MDA, a significant difference was observed in the knowledge about signs and symptoms of STH among the pupils in the intervention group compared to control group (p<0.001) (Table 6).

### Impact of Worms and Ladders game on the Attitude and Practices of the school children towards STH

Table 7 shows the difference between control and intervention group at the three different time points. There was no significant difference at baseline between the intervention and control groups for the hygiene attitude and practices observed. However, significant differences were observed at 3-months and 6-months follow-up period for all the hygiene attitude and practices observed except biting of fingernails (p<0.001). Findings from the focus group discussions among children also revealed the positive impact of Worms and Ladders game on attitude and practices.

**Table 5. Knowledge about Prevention and Control of STH among the school children.**

| Variables | Baseline | | | Follow-up (3 months) | | | Endline (6 months) | | |
|---|---|---|---|---|---|---|---|---|---|
| | Int n = 212 | Cont. n = 160 | p-value | Int n = 193 | Cont. n = 135 | p-value | Int n = 142 | Cont. n = 113 | p-value |
| **How can we prevent and control STH** | N (%) | N (%) | | N (%) | N (%) | | N (%) | N (%) | |
| Know at least one way of Prevention | 12 (5.7) | 15 (9.4) | 0.172 | 187 (96.9) | 19 (14.1) | <0.001 | 140 (98.6) | 16 (14.2) | <0.001 |
| Wash hand after defecation | 0 | 2 (1.2) | 0.103 | 63 (32.6) | 1 (0.7) | <0.001 | 58 (40.8) | 2 (1.8) | <0.001 |
| Wear slippers or shoes | 0 | 0 | - | 78 (40.4) | 0 | <0.001 | 77 (54.2) | 2 (1.8) | <0.001 |
| Use toilet | 0 | 0 | - | 35 (18.1) | 0 | <0.001 | 43 (30.3) | 0 | <0.001 |
| Wash fruits and vegetables before eating | 0 | 3 (1.9) | 0.045 | 28 (14.5) | 2 (1.5) | <0.001 | 29 (20.4) | 2 (1.8) | <0.001 |
| Wash hand before eating | 2 (0.9) | 2 (1.2) | 0.777 | 72 (37.3) | 2 (1.5) | <0.001 | 74 (52.1) | 3 (2.7) | <0.001 |
| Cut and clean fingernails | 0 | 1 (0.6) | 0.249 | 13 (6.7) | 1 (0.7) | 0.008 | 11 (7.7) | 2 (1.8) | 0.031 |
| Cover food from flies | 0 | 0 | - | 15 (7.8) | 0 | 0.001 | 15 (10.6) | 0 | <0.001 |
| Not biting fingernails | 0 | 0 | - | 7 (3.6) | 0 | 0.025 | 9 (6.3) | 0 | 0.006 |
| Not playing with soil | 1 (0.5) | 0 | 0.384 | 42 (21.8) | 1 (0.7) | <0.001 | 39 (27.5) | 3 (2.7) | <0.001 |
| Not picking food from ground | 0 | 0 | - | 7 (3.6) | 0 | 0.025 | 6 (4.2) | 0 | 0.027 |
| Availability of water-closet | 0 | 0 | - | 9 (4.7) | 0 | 0.011 | 7 (4.9) | 0 | 0.017 |

* Int- Intervention group, Cont.- Control group, N- Number

**Table 6. Knowledge about signs and symptoms of STH among the school children.**

| Variables | Baseline | | | Follow-up (3 months) | | | Endline (6 months) | | |
|---|---|---|---|---|---|---|---|---|---|
| | Int | Cont. | p- | Int | Cont. | p- | Int | Con | p- |
| | n = 212 | n = 160 | value | n = 193 | n = 135 | value | n = 142 | n = 113 | value |
| **Signs and Symptoms** | N (%) | N (%) | | N (%) | N (%) | | N (%) | N (%) | |
| Poor performance in school | 0 | 2 (1.2) | 0.103 | 14 (7.3) | 2 (1.5) | 0.017 | 11 (7.7) | 3 (2.7) | 0.076 |
| Stunted growth | 0 | 2 (1.2) | 0.103 | 38 (19.7) | 2 (1.5) | <0.000 | 25 (17.6) | 2 (1.8) | <0.001 |
| Blood loss | 0 | 0 | - | 46 (23.8) | 0 | <0.001 | 36 (25.4) | 0 | <0.001 |
| Fatigue | 5 (2.4) | 3 (1.9) | 0.750 | 44 (22.8) | 3 (2.2) | <0.001 | 30 (21.1) | 3 (12.9) | <0.001 |
| Abdominal pain | 5 (2.4) | 17 (10.6) | 0.001 | 36 (18.7) | 8 (5.9) | 0.001 | 26 (18.3) | 5 (4.4) | 0.001 |
| Rectal prolapsed | 0 | 0 | - | 6 (3.1) | 0 | 0.039 | 4 (2.8) | 0 | 0.072 |
| Swollen stomach | 0 | 4 (2.5) | 0.021 | 8 (4.1) | 2 (1.5) | 0.167 | 5 (3.5) | 2 (1.8) | 0.395 |
| Dysentery | 7 (3.3) | 5 (3.1) | 0.924 | 16 (8.3) | 8 (5.9) | 0.418 | 11 (7.7) | 7 (6.2) | 0.631 |
| Sickness | 0 | 0 | - | 2 (1.0) | 0 | 0.235 | 5 (3.5) | 0 | 0.044 |

* Int- Intervention group, Cont.- Control group, N- Number

"*I used to play with soil, but now that I know that one can become infected with worm through playing with soil, I don't play with soil again, in fact, I do not allow my siblings and my friends at home to play with soil*".

Male pupil at EAC Primary School Itoko

"*I now beat my siblings, and even shout on my elder brother if they walk barefooted, I do tell them worms will suck all their blood, and they will not have blood in their body again if they don't wear slippers/shoe outside*".

**Table 7. Impact of Worms and Ladders game on the Attitude and Practice of the school children towards STH.**

| | Int. n = 212 | Cont. n = 160 | p-value | Int. n = 193 | Cont. n = 135 | p-value | Int. n = 142 | Cont. n = 133 | p-value |
|---|---|---|---|---|---|---|---|---|---|
| | N(%) | N(%) | | N(%) | N(%) | | N(%) | N(%) | |
| **Wash hand before eating** | | | | | | | | | |
| No | 34(16.0) | 20(12.5) | 0.338 | 1(0.5) | 15(11.1) | <0.001 | 1(0.7) | 12(10.6) | <0.001 |
| Yes | 178(84.0) | 140(87.5) | | 192(99.5) | 120(88.9) | | 141(99.3) | 101(89.4) | |
| **Wash hands with** | | | | | | | | | |
| Nil | 35(16.5) | 20(12.5) | 0.256 | 1(0.5) | 15(11.1) | <0.001 | 1(0.7) | 12(10.6) | <0.001 |
| Water | 48(22.6) | 47(29.4) | | 5(2.6) | 41(30.4) | | 5(3.5) | 41(36.3) | |
| Water and soap | 129(60.8) | 93(58.1) | | 187(96.9) | 79(58.5) | | 136(95.8) | 60(53.1) | |
| **Bite fingernails** | | | | | | | | | |
| No | 153(72.2) | 106(66.2) | 0.219 | 152(78.8) | 88(65.2) | 0.005 | 110(77.5) | 82(72.6) | 0.368 |
| Yes | 59(27.8) | 54(33.8) | | 41(21.2) | 47(34.8) | | 32(22.5) | 31(27.4) | |
| **Fingernails cut and clean** | | | | | | | | | |
| No | 160(75.5) | 127(79.4) | 0.375 | 131(67.9) | 110(81.5) | 0.004 | 98(69.0) | 91(80.5) | 0.037 |
| Yes | 52(24.5) | 33(20.6) | | 62(32.1) | 25(18.5) | | 44(31.0) | 22(19.5) | |
| **Walk barefoot** | | | | | | | | | |
| No | 111(52.4) | 97(60.6) | 0.112 | 189(97.9) | 80(59.3) | <0.001 | 138(97.2) | 63(55.8) | <0.001 |
| Yes | 101(47.6) | 63(39.4) | | 4(2.1) | 55(40.7) | | 4(2.8) | 50(44.2) | |

* Int- Intervention group, Cont.- Control group, N- Number

Female pupil at EAC Primary School, Itoko

"*I used to walk barefooted, but now I always go out wearing my slippers.*"

Male pupil at Baptist Primary School Bode-Ijaiye,

"*I now make sure we wash our hands before eating at home.*"

Female pupil at St. John Primary School 1, Kuto

During one of the visits to St. John Primary School 1, Kuto, some children in nursery one class playing with soil outside their classroom, were admonished by their colleagues that "*worms will bite you, you are playing with soil*", and the children ran inside their classroom.

## Discussion

Efforts to prevent and control soil-transmitted helminthiasis require an integrated approach consisting of mass treatment campaigns with albendazole, and hygiene improvements achieved through access to improved and adequate water, sanitation and health education [13,28,35]. However, despite ongoing mass treatment campaigns in Ogun state, there is plethora of evidence on the endemicity of STH infections, and as well, high records of its prevalence among school children and pre-schoolers [10,26,27,35,36]. It is, therefore, necessary to complement the gains of mass treatment campaigns with other health interventions such as hygiene education. Hygiene education has been advocated as a reliable tool capable of interrupting STHs parasites transmission cycle. However, deploying an effective, sustainable health education messages to school-aged children within and outside the school system has been the challenge in STH control. In China, video games promoted significant gains in KAP aimed at controlling STH infections [13]. Conversely, a new health educational board game Worms and Ladder game, designed in this study, was tested to demonstrate its impact on reducing STH reinfection rates among school children in Abeokuta, Nigeria.

At first follow-up and second follow-up (i.e. three months and six months after deworming and introducing the game), children from intervention schools had lower levels of STH infection when compared with children from control schools. There was also a significant improvement in STH-related knowledge, attitudes and practices. These results suggest that the introduced health educational game was successful at improving school children's KAP regarding STHs and in turn, reducing STH reinfection rates [22–23, 37–38].

Furthermore, treatment with albendazole was effective at reducing the prevalence of STH parasites among the school children in both the intervention and control group following baseline assessments. However, an increase in prevalence was recorded among the control group, six months after treatment. In contrast, a significant reduction in prevalence was found among the children who received health education through the Worms and Ladders game in the intervention group. The upsurge in the prevalence of STH among children in the control group, therefore, reaffirms previous prepositions that, mass treatment campaigns as a standalone intervention, cannot prevent reinfections [13,22].

On the other hand, it is not uncommon that children may get re-infected following deworming once the environments are contaminated with viable eggs through open defecation. As such, infection intensity estimate, as measured by the number of parasitic eggs per gram of feces are more appropriate morbidity indicators than prevalence estimates at the population level. A decrease in intensity levels, therefore, represents reduced morbidity due to STH infections, and an indicator of essential health benefits [39]. On such premise, the findings from this study, therefore, suggest that improvement in knowledge through the

introduced game, translated into improvements in healthy hygiene behaviors, which led to a statistically significant reduction in the intensity of *Ascaris lumbricoides* infection in the intervention group. However, the intensity of hookworm infections was not statistically different among the intervention and control group after six months of exposure to the newly developed game. This is understandable as hookworm infections could be acquired through cutaneous penetration of the infective larvae, especially around the foot area, and only possession and usage of footwear while walking on contaminated soils can prevent such penetration. Majority of the children in the study area are not privileged to have more than a school sandal, which they protect by putting them off when playing soccer or on fields around the school premises [35]. It is thus evident that under such scenarios, hookworm infection is bound to persist even though the children are knowledgeable about the transmission of the disease. Health education has been previously reported to have a minimal, insignificant impact on the transmission of hookworm infections in the absence of footwears [22]. There was also no significant reduction in the intensity of *Trichuris trichiura* infection after six months post-deworming and health education intervention [22]. This is not outlandish, given that albendazole has low cure rates (28%) on *Trichuris trichiura* infection compared to 88% for *Ascaris lumbricoides* infection and 72% for hookworm infections [40].

Nevertheless, findings from this study corroborate with several studies showing health education is effective at reducing the rate of STH infections through increased knowledge about transmission of disease and improved hygiene attitude and practices [13,20,23,37,38]. However, complementing MDA with health educational tools have perceived challenges related to the high cost of production, delivery and adoption mechanisms to end-users, and as well as its affordability.

Comprehensively, the cost of production the Worms and Ladder game is $1.50 per unit. This cost is comparatively cheaper than other health educational tools such as board bills, flyers and posters, currently used during MDA campaign in African countries [41,42]. In terms of delivery and adoption, it could be burdensome for the parents to pay for the game, which could hinder the aim of controlling of STH infections. Considering the economic situation, the game could be mass-produced for MDAs by individuals and organizations after appropriate copyright transfer. Also, the game can be incorporated into the health education curriculum for schools in Nigeria to facilitate gains, with students encouraged to play during leisure periods. Alternatively, the game could be reproduced at the back of an exercise book for the school children, making the game easily accessible to the children even when at home. However, in areas with considerable access to internet facilities, the game can also be developed as a mobile application on smartphones users. We aim to scale up the evaluation of the game and test it against existing health education delivery channels for STH control.

## Conclusion

A new health education board game Worms and Ladders has been shown to improve the knowledge, attitude and practices of school-age children about transmission, prevention and control of STH infections. The game as an intervention tool significantly reduced reinfection with STHs after deworming at 3-months and 6-month post-intervention. This game, therefore, can be used as a behavioral change tool to enhance the impact of MDA for the control of STH infections in endemic settings, most especially if the goals of eliminating STH as a public health problem must be realized.

### Limitation of the study

Although we were unbiased in the selection of the study schools and participants, the lack of comparability in age distribution between the intervention and control groups, as observed,

can be attributed to the dynamics of voluntary participation and the need to obtained informed consent from the children and their parents. Unfortunately, we did not account for this in our analysis. Nevertheless, all study participants were from public primary schools, located in an area with similar socio-economic conditions.

## Supporting information

**S1 Data. Research data.**
(XLSX)

## Acknowledgments

We are grateful to the headteachers, staff and the pupils in the six primary schools for their support. Our profound gratitude goes to the Ogun State Ministry of Health, Ogun State Universal Basic Education Board and Abeokuta South Local Government Education Authority for their collaboration in this study.

## Author Contributions

**Conceptualization:** Dorcas B. Bassey, Akinola S. Oluwole, Uwem F. Ekpo.

**Data curation:** Dorcas B. Bassey, Hammed O. Mogaji, Abdulhakeem A. Adeniran, Olagunju A. Agboola.

**Formal analysis:** Dorcas B. Bassey, Hammed O. Mogaji.

**Investigation:** Dorcas B. Bassey, Gabriel A. Dedeke, Bolanle I. Akeredolu-Ale, Olagunju A. Agboola, Uwem F. Ekpo.

**Methodology:** Dorcas B. Bassey, Hammed O. Mogaji, Eniola M. Abe, Akinola S. Oluwole, Abdulhakeem A. Adeniran, Olagunju A. Agboola, Uwem F. Ekpo.

**Project administration:** Bolanle I. Akeredolu-Ale, Akinola S. Oluwole, Uwem F. Ekpo.

**Software:** Hammed O. Mogaji.

**Supervision:** Hammed O. Mogaji, Gabriel A. Dedeke, Bolanle I. Akeredolu-Ale, Eniola M. Abe, Akinola S. Oluwole, Abdulhakeem A. Adeniran, Chiedu F. Mafiana, Uwem F. Ekpo.

**Validation:** Bolanle I. Akeredolu-Ale, Eniola M. Abe, Akinola S. Oluwole, Abdulhakeem A. Adeniran, Chiedu F. Mafiana, Uwem F. Ekpo.

**Writing – original draft:** Hammed O. Mogaji.

**Writing – review & editing:** Dorcas B. Bassey, Gabriel A. Dedeke, Bolanle I. Akeredolu-Ale, Eniola M. Abe, Akinola S. Oluwole, Abdulhakeem A. Adeniran, Olagunju A. Agboola, Chiedu F. Mafiana, Uwem F. Ekpo.

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
