## [Decision Letter · Decision Letter 0]

17 Dec 2019

Dear Dr Mogaji:

Thank you very much for submitting your manuscript "The impact of “Worms and LaddersTM”, an innovative health educational board game on Soil Transmitted Helminthiasis control in Abeokuta, Southwest Nigeria" (#PNTD-D-19-01582) for review by PLOS Neglected Tropical Diseases. Your manuscript was fully evaluated at the editorial level and by independent peer reviewers. The reviewers appreciated the attention to an important problem, but raised some substantial concerns about the manuscript as it currently stands. These issues must be addressed before we would be willing to consider a revised version of your study. We cannot, of course, promise publication at that time.

We therefore ask you to modify the manuscript according to the review recommendations before we can consider your manuscript for acceptance. Your revisions should address the specific points made by each reviewer. 

When you are ready to resubmit, please be prepared to upload the following:

(1) A letter containing a detailed list of your responses to the review comments and a description of the changes you have made in the manuscript.

(2) Two versions of the manuscript: one with either highlights or tracked changes denoting where the text has been changed (uploaded as a "Revised Article with Changes Highlighted" file); the other a clean version (uploaded as the article file).

(3) If available, a striking still image (a new image if one is available or an existing one from within your manuscript). If your manuscript is accepted for publication, this image may be featured on our website. Images should ideally be high resolution, eye-catching, single panel images; where one is available, please use 'add file' at the time of resubmission and select 'striking image' as the file type. 

Please provide a short caption, including credits, uploaded as a separate "Other" file. If your image is from someone other than yourself, please ensure that the artist has read and agreed to the terms and conditions of the Creative Commons Attribution License at http://journals.plos.org/plosntds/s/content-license (NOTE: we cannot publish copyrighted images). 

(4) If applicable, we encourage you to add a list of accession numbers/ID numbers for genes and proteins mentioned in the text (these should be listed as a paragraph at the end of the manuscript). You can supply accession numbers for any database, so long as the database is publicly accessible and stable. Examples include LocusLink and SwissProt.

(5) To enhance the reproducibility of your results, we recommend that you deposit your laboratory protocols in protocols.io, where a protocol can be assigned its own identifier (DOI) such that it can be cited independently in the future. For instructions see http://journals.plos.org/plosntds/s/submission-guidelines#loc-methods

While revising your submission, please upload your figure files to the Preflight Analysis and Conversion Engine (PACE) digital diagnostic tool, https://pacev2.apexcovantage.com/ PACE helps ensure that figures meet PLOS requirements. To use PACE, you must first register as a user. Then, login and navigate to the UPLOAD tab, where you will find detailed instructions on how to use the tool. If you encounter any issues or have any questions when using PACE, please email us at figures@plos.org.

We hope to receive your revised manuscript by Feb 15 2020 11:59PM. If you anticipate any delay in its return, we ask that you let us know the expected resubmission date by replying to this email.

To submit a revision, go to https://www.editorialmanager.com/pntd/ and log in as an Author. You will see a menu item call Submission Needing Revision. You will find your submission record there. 

Sincerely,

W. Evan Secor

Associate Editor

Mar Siles-Lucas

Deputy Editor

Reviewer's Responses to Questions

**Key Review Criteria Required for Acceptance?**

**Methods**

-Are the objectives of the study clearly articulated with a clear testable hypothesis stated?

-Is the study design appropriate to address the stated objectives?

-Is the population clearly described and appropriate for the hypothesis being tested?

-Is the sample size sufficient to ensure adequate power to address the hypothesis being tested?

-Were correct statistical analysis used to support conclusions?

-Are there concerns about ethical or regulatory requirements being met?

Reviewer #1: - The objectives were stated.

- The study design is appropriate.

- The assignment procedure of treatment groups is not so clear for me. Did you assign the schools to the treatment groups by each cluster that you made during the selection procedure?

- How many days did you collect fecal samples at the same timeline?

- How did you ask the KAP questionnaires to the participants? (Self-administered or interview?) Which language did you use?

- At baseline survey, did you give treatment to all participants or those who were positive of STHs?

- Did you give treatment at follow-up and endline surveys?

- Do you have an ethical approval number?

- Almost half of the participants are age > 10 years old. Did you obtain assent from these participants?

Reviewer #2: - Line 111-112, if I understood well, the authors are referring the game as “treatment”. I would suggest to name it “intervention”. 

- After baseline, children were dewormed with a single dose of albendazole. Given that a single dose could be underdosing for treatment, the authors do not mentioned the collection of a second stool for the positives at baseline to ensure cure. 

- Line 163, do not specify what follow- up consists, a sample collection, questionnaire… 

- In lines 175-180 it is not specified if the children new about the recording. They should be informed for ethical reasons. If so, the discussion collected is susceptible to be biased. This and the KAP procedure is a bit unclear. 

- No explanation on characteristics of the 6 studied schools (e.g. number of students, water, sanitation and hygiene conditions at the community and at the school…) to ensure they are comparable and reinfection is not due to other factors.

Reviewer #3: The description of the intervention is quite basic. A more complete description of the messages used, as well as the auxiliary discussion with teachers is warranted.

For the intervention, I would like to see some discussion about the merits of knowledge-based education as a driver of behavior change. I am surprised that knowledge alone would be sufficient to get children to change their behavior without, for example, improvements in access to water and soap or demonstrations of proper HW practice. Perhaps the authors need to provide a bit more context for the study site, since this approach may not work in areas that are more resource/water scarce.

There is no sample size calculation included. 

There is no description of how students were selected within the schools and the types of surveys that were conducted and how (there is a quote from a student in line 345).

Was the analysis conducted to account for clustering at a school-level?

How were costs calculated? Line 436 points to the cost of the game, but what are the other costs associated with the intervention (e.g., teacher time)

**Results**

-Does the analysis presented match the analysis plan?

-Are the results clearly and completely presented?

-Are the figures (Tables, Images) of sufficient quality for clarity?

Reviewer #1: - Please show that the control and intervention groups are comparable. I think the prevalence of STHs and age distribution at baseline are too different to compare.

- When you describe the result which is not statistically significant, it is not necessary to write p-value > 0.05 in the text. (Line 263, 289, 312)

- When you describe the result which is statistically significant, please show the exact p-value. (Line 226, 251, 344).

- Line 342-346 You may need to change the phrases. Because table 6 shows the comparison of control and intervention groups at baseline, follow-up ,and endline but not comparison within each treatment group. So based on the results in table 6, you can say the difference between control and intervention groups at 3-time points but not the difference within the group over time.

- Do you have any results of group discussion? (line 345-346 only?)

- Only table 2 uses “group 1” and “group 2”. Please change to control and intervention.

Reviewer #2: - Lines 193-195 and table 1, 56.9% (intervention) and 49.1% (control) do not sum 100%. Not sure there is an error. 

- Figure 4-6, I would add “(%)” after “Prevalence”.

Reviewer #3: The analysis approach is not clear, thus the results are not clearly described. I would expect that the main outcome is a risk double difference in the prevalence STH at endline comparing intervention to control, accounting for baseline values. In that way, both time points could be used in the analysis. As it is currently shown, it seems that p values (but no estimates of effect or CIs) are just comparing intervention to control for each round (including baseline)?

Along those lines, it is not appropriate to only report p values. Risk ratios or odds ratios with 95% CI would be more appropriate. Reliance on p values alone is insufficient for the reader to understand variance.

The figures (charts) are not particularly useful and would be better represented in tables.

Some of the text can be cut in lieu of putting data in the tables.

**Conclusions**

-Are the conclusions supported by the data presented?

-Are the limitations of analysis clearly described?

-Do the authors discuss how these data can be helpful to advance our understanding of the topic under study?

-Is public health relevance addressed?

Reviewer #1: - No limitation was described.

- It may need much more concrete recommendations to the public health sector.

Reviewer #2: They are very accurate to the results and analysis they got. 

They should mention the limitations of not considering water, sanitation and hygiene community and shcool conditions. 

They also should mentioned the possibility of the children uncured, since they do not mention to monitor that. 

Lines 165-173 , it is not well understood if schoolchildren were forced to play the game during leisure time or it was volunteer,

Reviewer #3: The conclusions should better address the context as discussed above.

More detail on the implications for the use of games and knowledge-based educational campaigns to impact hygiene behavior (and what behaviors were targeted) would be warranted.

**Editorial and Data Presentation Modifications?**

Reviewer #1: (No Response)

Reviewer #2: Lines 165 to 180 are a little unclear, some detail is missing.

Reviewer #3: There is some need for grammar and english language editing to enhance clarity.

**Summary and General Comments**

Reviewer #1: This paper showed the results of a randomized control trial (RCT) to examine the impact of an educational board game on STHs control. The authors found the improvement of KAP and the suppression of STHs reinfection among school-aged children after the introduction of an educational board game. However, at the baseline, they did not show whether the control and the intervention groups are comparative or not. For publication, it needs to show that those two groups are comparable, and if they have a statistical difference at baseline, it is necessary to explain how the difference appeared and influenced the results.

About the board game development, some board games similar to “Worms and Ladders” have been already developed and it is available on several websites.

https://www.who.int/intestinal_worms/resources/Global_materials/en/

https://www.who.int/intestinal_worms/resources/WACIPAC_worms_ladders_game_Eng.pdf?ua=1

https://www.who.int/intestinal_worms/resources/WACIPAC_Worms_Ladders_instructions_Eng.pdf

https://unitingtocombatntds.org/womendeliver/worms-and-ladders/

If the authors referred to those games, please mention them in the text.

Reviewer #2: The main idea of the study is great and necessary. It seems to be a good approach to reduce soil-transmitted helminths reinfection. However, the methodology has some limitations.

Reviewer #3: With RCTs, it is common to submit pre-analysis plan to a database such as CinicalTrials.org prior to the study. Was this done?

Is this game really trade marked, as indicated by the name Worms and Ladders"TM"? If so, who owns the trademark and is this something that needs to be disclosed as a conflict of interest?

How was the game developed? Was it piloted? How were messages decided upon? Did the researchers rely on any behavioral theory in developing the intervention?

In the introduction, more discussion is needed on the evidence of education on behavior change related to hygiene in general, and STH prevention specifically. Additional literature on the use of games for behavior change would be useful.

PLOS authors have the option to publish the peer review history of their article (what does this mean?). If published, this will include your full peer review and any attached files.

Reviewer #1: No

Reviewer #2: No

Reviewer #3: No

---

## [Decision Letter · Decision Letter 1]

8 May 2020

Dear Dr Mogaji,

Thank you very much for submitting your manuscript "The impact of “Worms and LaddersTM”, an innovative health educational board game on Soil Transmitted Helminthiasis control in Abeokuta, Southwest Nigeria" for consideration at PLOS Neglected Tropical Diseases. As with all papers reviewed by the journal, your manuscript was reviewed by members of the editorial board and by several independent reviewers. The reviewers appreciated the attention to an important topic. Based on the reviews, we are likely to accept this manuscript for publication, providing that you modify the manuscript according to the review recommendations. 

Please address the additional comments of the reviewers. Also, please pay close attention to the comments of reviewer 2 on the original submission to make sure they have been addressed.

Sincerely,

W. Evan Secor

Associate Editor

Mar Siles-Lucas

Deputy Editor

Please address the additional comments of the reviewers. Also, please pay close attention to the comments of reviewer 2 on the original submission to make sure they have been addressed.

Reviewer's Responses to Questions

**Key Review Criteria Required for Acceptance?**

**Methods**

-Are the objectives of the study clearly articulated with a clear testable hypothesis stated?

-Is the study design appropriate to address the stated objectives?

-Is the population clearly described and appropriate for the hypothesis being tested?

-Is the sample size sufficient to ensure adequate power to address the hypothesis being tested?

-Were correct statistical analysis used to support conclusions?

-Are there concerns about ethical or regulatory requirements being met?

Reviewer #1: - The assignment procedure of treatment group is still not clear for me. In the text, you wrote “The selected schools were blindly assigned to two treatments; Treatment 1 (“Worms and LaddersTM” game); and Treatment 2 (“Snakes and Ladders” game) by another researcher who did not partake in the selection procedures” (line 132-134). However, Figure 1 shows that the schools from cluster 1 go to intervention group and schools from cluster 2 go to control group.

Reviewer #2: It is my second revision of this manuscript. My comments on this version methods are:

- I suggest to change the term for the game from "treatment" to "intervention". 

- The authors states that schools were “blindly assigned” but not sure if it was systemically or randomly assigned. 

- Lines 180, I am not sure if authors were meaning that an assent form was completed by children older than 16 years old. 

- They calculate odds ratio but they do not adjust for confounders that could be interferring the risk.

**Results**

-Does the analysis presented match the analysis plan?

-Are the results clearly and completely presented?

-Are the figures (Tables, Images) of sufficient quality for clarity?

Reviewer #1: - The p-values in table 2, 4, 5, 6, 7: Are they p-values for comparison of intervention and control groups at each time point? If so, you cannot describe the longitudinal change within groups (line 319-320, 328-330, 333-334, 348-349, 397-399). If you would like to show the longitudinal change, p-values for comparison base-line vs follow-up/end-line are needed.

- Table 2 and 3: It may be better to combine two tables. It is NOT necessary to show both OR and RR. It is better to choose the one of them (OR or RR) to show.

Reviewer #2: - Prevalence is double in the control group, so they are not comparable.

**Conclusions**

-Are the conclusions supported by the data presented?

-Are the limitations of analysis clearly described?

-Do the authors discuss how these data can be helpful to advance our understanding of the topic under study?

-Is public health relevance addressed?

Reviewer #1: (No Response)

Reviewer #2: -

**Editorial and Data Presentation Modifications?**

Reviewer #1: (No Response)

Reviewer #2: -

**Summary and General Comments**

Reviewer #1: (No Response)

Reviewer #2: Some of the comments were already done in the last revision but were not changed.

PLOS authors have the option to publish the peer review history of their article (what does this mean?). If published, this will include your full peer review and any attached files.

Reviewer #1: No

Reviewer #2: No
---

## [Decision Letter · Decision Letter 2]

15 Jun 2020

Dear Dr Mogaji,

We are pleased to inform you that your manuscript 'The impact of Worms and Ladders, an innovative health educational board game on Soil Transmitted Helminthiasis control in Abeokuta, Southwest Nigeria' has been provisionally accepted for publication in PLOS Neglected Tropical Diseases.

Best regards,

W. Evan Secor

Associate Editor

Mar Siles-Lucas

Deputy Editor

The paper is acceptable for publication but please make the revisions requested by Reviewer 1 in the final version. In addition please make clear in the sentence on line 39 that the 5.6% prevalence value refers to the sixth month time point. Currently, it is not clear when this measurement was made. Also, a "reliable" is a better word choice than "veritable" in lines 45, 60, and 479. In line 85, changing the word "more" to "longer" makes better sense for the thought the authors are trying to convey.

Reviewer's Responses to Questions

**Key Review Criteria Required for Acceptance?**

**Methods**

-Are the objectives of the study clearly articulated with a clear testable hypothesis stated?

-Is the study design appropriate to address the stated objectives?

-Is the population clearly described and appropriate for the hypothesis being tested?

-Is the sample size sufficient to ensure adequate power to address the hypothesis being tested?

-Were correct statistical analysis used to support conclusions?

-Are there concerns about ethical or regulatory requirements being met?

Reviewer #1: - The reason that you assigned intervention/control by cluster was clear now (for me). However, your explanation on it in the main text (line 124-127) may be misplaced. Moreover, if you assigned intervention/control to each cluster, please explain how and when you assigned it.

- Please mention which model you used to analyse the longitudinal data.

Reviewer #2: All my previous comments and suggestions have been adressed.

**Results**

-Does the analysis presented match the analysis plan?

-Are the results clearly and completely presented?

-Are the figures (Tables, Images) of sufficient quality for clarity?

Reviewer #1: Line 313-314: There was a significant difference in post-intervention prevalence between the two groups (p=0.007).

Line 330-331: There was no significant difference in the reduction observed between the intervention and control group (p=0.898).

- Both text above and p-values showed in the text are inconsistent. The text describe the difference between the two groups, but the showing p-values are for longitudinal analysis according to your information.

- Line 284, line 325: odd  odds

Reviewer #2: All my previous comments and suggestions have been adressed.

**Conclusions**

-Are the conclusions supported by the data presented?

-Are the limitations of analysis clearly described?

-Do the authors discuss how these data can be helpful to advance our understanding of the topic under study?

-Is public health relevance addressed?

Reviewer #1: - As the other reviewer mentioned (I also pointed it out at the first round), the prevalence at baseline in each group were so different. We do not know why it was so different. It may be because the age distribution was different (it was already mentioned in the limitation.), there is different environmental factor in each cluster, only cluster 1 has got MDA before this research, or something else. Please mention it clearly as limitation.

Reviewer #2: All my previous comments and suggestions have been adressed.

**Editorial and Data Presentation Modifications?**

Reviewer #1: (No Response)

Reviewer #2: (No Response)

**Summary and General Comments**

Reviewer #1: (No Response)

Reviewer #2: All my previous comments and suggestions have been adressed.

PLOS authors have the option to publish the peer review history of their article (what does this mean?). If published, this will include your full peer review and any attached files.

Reviewer #1: No

Reviewer #2: No

---

## [Editor Report · Acceptance letter]

18 Sep 2020

Dear Dr Mogaji,

We are delighted to inform you that your manuscript, "The impact of Worms and Ladders, an innovative health educational board game on Soil Transmitted Helminthiasis control in Abeokuta, Southwest Nigeria," has been formally accepted for publication in PLOS Neglected Tropical Diseases.

Best regards,

Shaden Kamhawi

co-Editor-in-Chief

Paul Brindley

co-Editor-in-Chief
